# Raw Cow’s Milk and Its Protective Effect on Allergies and Asthma

**DOI:** 10.3390/nu11020469

**Published:** 2019-02-22

**Authors:** Barbara Sozańska

**Affiliations:** 1st Department of Pediatrics, Allergology and Cardiology, Wroclaw Medical University, Wroclaw 50-367, Poland; bsoz@go2.pl; Tel.: +48605370686

**Keywords:** raw milk, allergy, asthma, protection

## Abstract

Living on a farm and having contact with rural exposures have been proposed as one of the most promising ways to be protected against allergy and asthma development. There is a significant body of epidemiological evidence that consumption of raw milk in childhood and adulthood in farm but also nonfarm populations can be one of the most effective protective factors. The observation is even more intriguing when considering the fact that milk is one of the most common food allergens in childhood. The exact mechanisms underlying this association are still not well understood, but the role of raw milk ingredients such as proteins, fat and fatty acids, and bacterial components has been recently studied and its influence on the immune function has been documented. In this review, we present the current understanding of the protective effect of raw milk on allergies and asthma.

## 1. Introduction

Cow’s milk and its preserves are basic ingredients in our diet. For centuries it was introduced as the first infant food as an alternative to breast milk and treated as necessary for growth and development. Milk is now processed on an industrial scale to avoid the risk of pathogenic bacteria infection from unpasteurized milk. The consumption of raw milk is much less common than in the past, but a considerable number of consumers, especially living on farms, still drink milk directly from cows [1].

In the last few decades, a significant increase in the prevalence of allergic diseases and asthma has been observed. For over a decade, an accumulating body of evidence has indicated that living on a farm can reduce the risk of allergen sensitization and allergic diseases in children. Among the first observations were those in alpine villages where children of full-time farming parents had lower rates of atopy and hay fever than their peers living in the same rural community [2]. Further studies from the region, and in several other parts of the world, confirmed a protective effect of an early farming environment in children [3,4,5]. Findings have not been entirely consistent [6,7], suggesting that protective effects may depend on a particular type of farming environment. Although exposures in the first years of life are believed to exert the most important influence, and farming protective effects are best documented in children, it has been proposed that they may persist into adult life, and a lower prevalence of atopic sensitization among adult farmers has been observed [8,9,10]. Considerable effort has been put into deciphering which particular aspect(s) of a farm environment exert the protective effect. Success has so far been very limited, probably because it has proved impossible to separate individual exposures. Most proposed explanations have been based on variations in the “hygiene hypothesis” and a possible immunomodulatory effect of a farm environment. Specific types of exposure have been proposed, such as contact with farm animals, exposure to barns and animal sheds, and the diversity of microorganisms in farm environments [5,11,12]. One of the best documented protective effects on allergies and asthma is the consumption of unpasteurized milk products both in farming and non-farming environments [13]. The observation is even more intriguing, considering the fact that milk is one of the most common food allergens in childhood.

## 2. The Protective Effect of Raw Milk on Allergies in Farming and Non-Farming Populations—Epidemiological Studies

There are many cross-sectional surveys that confirm the protective effect of raw milk on allergies and asthma. First epidemiological evidence came from alpine regions. The ALEX study reported that children at school age were less likely to suffer from asthma or atopic sensitization when they consumed unpasteurized milk from the farm in their first year of life. This association was independent of living on a farm, although additional exposure to farm stables strengthened the effect [3]. These observations were confirmed in the next alpine study by Waser et al., where a significant inverse association with doctor diagnosed asthma, diagnosed rhinoconjunctivitis, and atopy was observed with respect to the consumption of farm milk in children from farming and non-farming families. The effect was independent of concomitant exposures to microbial compounds present in animal sheds and farm homes [14]. The protective effect of current farm milk consumption on atopy and eczema, but not asthma, in English children living in a rural environment was also observed [15]. In Crete, the consumption of unpasteurized milk has proven to protect against atopy, independently of having contact with farm animals and being a rural child [16]. In Poland, the consumption of raw milk in the first year of life reduced atopy risk even in adulthood, even more so in children, independent of where they resided. Current milk consumption also provided protection, albeit weaker. Protective effects on asthma were observed both in village and town inhabitants and at all ages, but for doctor-diagnosed hay fever the effect was only present for nonfarmers and those living in a town [17]. In a study of older adults in a US farming cohort, the consumption of raw milk was not associated with atopy and asthma, but raw milk as the primary milk consumed in childhood reduced odds of atopy [10].

In GABRIELA, the first study designed to find the biological components of cow’s milk that might explain the protective effect of farm milk on asthma and atopy in children, the reported consumption of raw milk in early life and currently was inversely related to atopy, hay fever, and asthma in school-aged children [18]. Heated farm milk was not associated with asthma outcomes. Total fat and protein content, total bacterial counts, and lactose levels had no association with allergies, atopy, or asthma. Inverse associations with asthma were found for α-lactalbumin, β-lactoglobulin, and bovine serum albumin (BSA) whey protein, but not with atopy. Although higher counts of microbes were detected in raw milk compared with pasteurized milk or heated farm milk, surprisingly there was no association between total bacterial counts and health outcomes. The cross-sectional design of the survey could not determine the long-term bacterial exposure due to raw milk consumption or how it could influence the gut microflora and immune system.

There are few publications dealing with the protective effect of raw milk on allergies and asthma. Most concern a birth cohort study, Protection against Allergy Study in Rural Environments (PASTURE), which included over 900 children from rural areas in five European countries [19]. Regular consumption of unpasteurized milk was inversely related to asthma onset at 6 years of age. This protective effect was stronger with recent exposure and in higher fat content milk. Higher ω-3 polyunsaturated fatty acids (PUFA) levels in milk was protective [20]. In the same cohort, maternal consumption of butter and unskimmed cow’s milk during pregnancy affected the fetal immune system for the production of interferon (INF)-gamma [21]. Interestingly, early consumption of raw farm milk not only protected from allergies and asthma but also exerted strong protection against rhinitis, otitis, and other respiratory track infections. This effect was strongest in raw milk. Boiled milk exhibited an attenuated effect [22]. In a recent study in the PASTURE cohort, cheese consumption and its diversity at 18 months of age had a protective effect on atopic eczema and food allergy but not on atopy or asthma at age 6. The authors proposed two possible explanations: (1) a positive effect of the microbial diversity of cheeses as a fermented food and its influence on the rich diversity of gut microbiota or (2) a potential anti-inflammatory effect by the inhibition of proinflammatory cytokines and intestinal microbiota metabolite production after dairy product consumption [23].

## 3. Raw Milk Versus Commercial Milk—What Are the Differences?

The epidemiological evidence showed that the protective effect on allergies and asthma is exerted by raw milk, but not (or weaker) by processed or commercial milk. The commercial milk differs in many aspects from raw farm milk, as the processing of milk changes its composition.

Raw milk is subjected to two basic processes: UHT homogenization and sterilization, which affect the content and functionality of fats, bacteria, and proteins in milk. Homogenization is the breaking of fat pellets into very small ones to prevent the formation of cream on the milk surface. This process changes the physical structure of fats but also casein and whey proteins. Splitting large spheres of fat into many small ones increases the total surface on which casein proteins are more easily adsorbed. In an experimental study on sensitized mice, only homogenized milk induced an allergic reaction in the intestinal wall of the animals, indicating that such milk processing may predispose one to an allergic reaction [24]. The quality of fat in milk depends on the ingredients cows are fed—if it is only grass, without any industrial mixtures, the content of polyunsaturated omega-3 fatty acids is higher. A prospective study by KOALA showed that the higher concentration of these acids in the milk of breastfeeding mothers reduced the risk of eczema at 2 years of age and 1 year of allergy in children [25].

Heating is another milk processing procedure. It might be the pasteurization process (heating up to about 75° for about 30 s) or the industrial, UHT sterilization (heating up to 130–160° and then quickly cooling), which reduces the bacterial content and enzymes that allow such milk to be stored without a refrigerator for several months.

Epidemiological evidence confirms that heating raw milk influences its allergy protective properties [18]. Although the main aim of heating is to reduce bacterial content and slow microbial growth, it may also affect the heat-sensitive milk components. Whey proteins are the most sensitive to this process. They lose their biological functionality by denaturation, aggregation, and glycation after heating [26]. Pasteurization at 72 °C denatures only part of the bovine IgG, but sterilization and homogenization denatures all immunoglobulins. Short heating up to 72 °C can change the lactoferrin structure and decrease its level in milk. Bovine transforming growth factor (TGF)-β1 concentrations were decreased in commercial pasteurized milk compared to raw milk [27]. The level of TGF-β2 did not differ between pasteurized and raw milk, but with increasing temperature a reduction of TGF-β2 concentrations in both was observed [28]. In the murine model of house dust mites (HDM) induced asthma, the effects of raw and heated raw cow’s milk on asthma prevention were compared. Airway hyperresponsiveness was prevented by raw milk but not heated milk. Raw milk reduced the total number of inflammatory cells such as eosinophils, lymphocytes, neutrophils, and macrophages in bronchoalveolar lavage fluid (BALF). Th2 and Th17 cells and their cytokine production of Interleukin (IL)-4 and IL-13 were reduced in lung cell suspensions both by raw and heated milk. Raw milk reduced IL-5 and IL-13 production after ex vivo restimulation of lung T cells with HDM [29]. These associations may confirm a causal relationship between raw cow’s milk consumption and the prevention of allergic asthma.

## 4. Raw Milk Proteins

The main fractions of milk proteins are casein (82% of total protein content), serum, and whey proteins (18% of total protein content). Casein is thermostable and is not destroyed during industrial processing. Whey protein is a group of a dozen proteins, including the major β-lactoglobulin milk allergen, α-lactalbumin, serum albumin, as well as immunoglobulins, lactoferrin, enzymes, and cytokines such as TGF-β and IL-10. Heating can change the physicochemical characteristics of these proteins and affect their biological impact. They are bioactive compounds that can influence the immunological system and prevent allergic reaction. The role of bioactive whey proteins from raw milk in allergic diseases is presented in detail in a recently published review [30]. The first study, which documented the effect of whey protein’s protective properties, was a Gabriela survey conducted on rural farm children [18]. The inverse associations with asthma were found for α-lactalbumin, β-lactoglobulin, BSA whey protein content, but not with atopy. Total protein content had no association with allergies, atopy, or asthma.

The caseins, β-lactoglobulin, and α-lactalbumin cannot be linked directly to immune functioning [31]. Other cow’s milk protein components such as lactoferrin, immunoglobulins, lysozyme, and cytokines (TGF-β, IL-10) may have a direct immunity-related effect.

Lactoferrin is an immunostimulator and immunoregulator of antigen presenting cells in culture [32]. It has inhibited the cytokine production of Th1 but not Th2 cell lines [33]. It has an antimicrobial effect (via direct interaction with the bacteria cell wall or by the binding iron needed by some of bacteria for growth), by which it can modulate microbial composition [31,34]. Bacteria with low iron requirements such as *Bifidobacteria* and *Lactobacilli* are promoted in the gut [35]. Their presence in an infant’s gut has been correlated with protection against allergies [36]. Moreover, lactoferrin reduced allergen-induced airway inflammation in a murine asthma model [37]. Its presence stimulated the production of TGF-β and IL-10 in the gut [38].

TGF-β is a multifunctional cytokine, of which higher levels (mainly TGF-β2 and TGF-β1 isoforms) were found in raw milk and in human breast milk of mothers from farm environments [27]. It plays a key role in the development and maturation of the mucosal immune system [39]. TGF-β1 increases the expression of intestinal tight junctions, which enhances barrier function of the gut [40]. This may potentially protect against food allergic sensitization and reduce allergy-related symptoms in infancy [39]. It has been shown in an animal model that this cytokine present in breast milk induced oral tolerance to allergens and protected from allergic asthma [41]. The lack of TGF-β in the milk formula promoted the production of the proinflammatory cytokine profile and increased numbers of eosinophils, activated mast cells, and dendritic cells in gut in another animal model. Supplementation of TGF-β induced oral tolerance to β-lactoglobulin from cow’s milk and increased IL-10 production [42]. IL-10 is a regulatory cytokine present in bovine milk. It inhibited immunoglobulin E (IgE) induced mast cell activation, Th2 cell activation, and eosinophil function [43]. The inverse correlation between IL-10 and allergic diseases and asthma were observed [44]. TGF-β and IL-10 induced conversion of naive peripheral T cells into FoxP3 regulatory T cells [45].

As was proposed by Neerven et al. [31], the consumption of immunomodulatory cytokines (such as TGF-β and IL-10) in unprocessed bovine milk may create the environment, which promotes regulatory T cell production, needed for developing and maintaining oral tolerance in the gut with IgA and IgG4, but not IgE production. IgA present in intestinal secretion can prevent binding food allergens to IgE. Low levels of human milk IgA correlated with allergy development [46]. To date, the role of immunoglobulins present in bovine milk (predominantly IgG) in allergy prevention is not well understood. We can speculate about the role of IgG in breast milk in forming immune complexes with allergens, but it has never been studied [47]. However, bovine milk may contain IgG antibodies specific for human allergens [31]. Theoretically they might suppress allergic responses by blocking Ig-E mediated activation of mast cells and basophils [30]. Recently, it has been shown that bovine IgG and raw milk can induce an innate immune memory in human monocytes modulating the responsiveness of the innate immune system to pathogen-related stimuli [48].

## 5. Fat and Fatty Acids

Milk fat content depends on the type of feeding and on the age and breed of cows, and it is a precious source of saturated and mono- and polyunsaturated fatty acids. Fat separation for adjusting fat milk levels and homogenization for fat creaming prevention are the main fat-changing processes in commercial milk production. The protective effect of high fat containing products such as full cream milk and butter on asthma was presented in the PASTURE cohort [14]. A higher content of anti-inflammatory ω-3 fatty acids in raw milk was found to be protective. The fat content was associated with asthma severity: high fat milk exerted a stronger protective effect on milder forms of asthma. Authors suggested that ω-3 PUFA may exert its effect by shifting the metabolic balance of eicosanoid synthesis from proinflammatory to anti-inflammatory mediators [20]. Short-chain fatty acids (SCFA) are metabolites present in relatively high concentration in bovine milk (not present in human milk fat), but may also be produced by microbes in the gut following the fermentation of fibers [31]. They may exert an anti-inflammatory effect by the inhibition of histone deacetylation, which influences the expansion of regulatory T cells, and may increase the production of IL-10 [49,50]. In a recent study in the PASTURE cohort, yogurt introduction in the first year of life increased the level of butyrate SCFA in fecal samples at one year of age, and the children with a high level of butyrate or propionate were protected from asthma and food allergy later in life and against atopy at age 6 [51]. Oral administration of butyrate, propionate, and acetate in mice experimental models reduced airway hyperresponsiveness during metacholine challenge and the number of inflammatory cells such as eosinophils in bronchoalveolar lavages [51]. Similarly, in a study among adults, the fecal butyrate was increased after yogurt consumption [52]. In a recent prospective observational study in an urban population, the inverse association between yogurt consumption in the first year of life and atopic eczema and food sensitization at five years of age was shown [53]. In another prospective study in New Zealand, a similar effect was observed at 12 months of age [54]. Authors speculate that the protective effect might be connected with probiotic bacteria in yogurt.

## 6. Microbial Content in Raw Milk

The presence and composition of bacteria is the clearest difference between unpasteurized and processed milk. Data from British laboratories presented by Perkin documented a greater total number of bacteria (including *Escherichia coli* and coagulase-positive *staphylococci)* in unpasteurized milk. Non-pathogenic *Listeria* species were found in 37% of raw milk samples and only in 0.4% of pasteurized samples [55]. The prevalence of pathogens in milk depends on numerous factors, but raw milk can be contaminated with pathogens even coming from healthy animals, as dairy farms may be the reservoir of various foodborne pathogens [1].

Surprisingly, the levels of lipopolysaccharides (LPS) endotoxins measured in raw and commercial milk did not differ in the PASTURE study [56]. On the contrary, in another study, the level of endotoxins was much higher in the samples of whole raw milk than in the processed shop milk and cold storage or heating increased the endotoxin concentrations in farm but not in the processed milk [57]. In the GABRIELA study, viable bacterial cell counts were higher in raw milk than in shop milk, but these differences were not associated with asthma and atopy [18]. In a recent study by Brick et al., the presence of the CD14 molecule in raw milk, a receptor of bacterial endotoxin, was confirmed. [26]. Such bacterial endotoxins present in farm dust modified the mechanisms of primarily non-specific immune response to allergens. House dust mites induce the activation of airway epithelial cells mediated by toll-like receptor 4. Airway exposure to endotoxins inhibited activation of NF-ĸB by the increase in the synthesis of its attenuator, enzyme A20. These associations, observed in an experimental model in mice, have been confirmed in further experiments on human bronchial epithelial cultures and in a case–control study of asthmatics [58]. In other studies, cluster of differentiation 14 (CD14) receptor polymorphisms at exposure to bacterial endotoxins in the first year of life in children from rural households reduced the risk of developing bronchial asthma [59] and atopic eczema [60].

Bieli et al. investigated whether the CD14 receptor polymorphism for bacterial endotoxins modifies the appearance of a protective effect on the consumption of unpasteurized milk in asthma and hay fever in children. Polymorphisms of the promoter gene for CD14 (CD14/-1721) had an effect on the occurrence of a protective effect. The inverse relationship between drinking unpasteurized milk and the occurrence of asthma, allergic rhinoconjunctivitis, and wheezing was most strongly expressed in the homozygote AA (adenine-adenine), less expressed in heterozygous AG (adenine-guanine), and did not occur in children with the genotype GG (guanine-guanine). This effect was independent of the children’s place of residence (rural farm or not). The authors speculate that the effect on the immune system through CD14 may proceed either due to the microbiological components of unpasteurized milk or intestinal microflora altered under the influence of probiotics delivered with milk. Alternatively, as it is known that CD14 is also a phospholipid receptor, fatty components of milk, including omega-3 fatty acids, can also play a role [61].

In the experimental murine model of gastrointestinal allergy, feeding with the untreated raw milk containing bacteria induced a greater allergic response (measured by specific IgE antibody level and MMCP-1) than with sterilized milk and heated milk. A higher in vitro production of IL-10 by splenocytes, regulatory cytokine playing an important role in suppressing allergy responses, was also found [28].

The protective properties of raw milk against allergies and asthma may be connected not only with the direct content of bacteria but also with the influence of different raw milk ingredients on gut microbiome. Microbial gut composition may be associated with increased risk for atopy, eczema, or wheeze, and it may differ depending on allergy status [62,63]. Early gut microbiome composition may also be associated with allergy resolution later in life, as was shown in a study of milk-allergic children [64]. Some factors present in raw milk may potentially modulate microbiota composition. Proteins such as lactoferrin and lysosyme have antimicrobial activity. Saccharides can promote the growth of bifidobacteria. This may influence the production of SCFAs, such as acetate, butyrate, and propionate, and enhance the epithelial barrier function of the gut [31]. Moreover, it was shown that short chain fatty acids affected the bone marrow dendritic cell maturation and inhibited the Th2-dependent response, which reduced allergic airway inflammation after allergen exposure. Decreased fatty acid metabolism by gut microbiota was associated with milk allergy resolution with time [64].

## 7. Milk Exosomes

Bovine and human milk contain substantial amounts of exosomal miRNAs (miRs) that may be transferred to the infant to promote immune regulatory functions. Raw cow’s milk contains high amounts of bioactive miRs. Pasteurization process decreases its level. Boiling milk results in complete miRs degradation. Milk exosomes are of critical importance for the maturation of the immune system during the postnatal period and early infancy. Milk-derived exosomal microRNAs may be potential stimuli for thymic Treg maturation and raw milk-mediated atopy prevention [65]. Farm milk exposure has been associated with increased numbers of CD4+CD25+FoxP3+ regulatory T cells (Tregs), lower atopic sensitization, and asthma in 4.5-year-old children [66]. Milk miRs may promote a selection process turning self-reactive thymocytes into stable Treg cells, and functionally active FoxP3 Treg cells suppress the development of Th2 cell-dependent immune responses. It has been suggested that milk’s exosomal miR system may represent “the missing candidate” inducing atopy-preventive effects of raw cow’s milk consumption, and the future prevention of atopic diseases might be possible by an addition of appropriate miR-155-enriched exosomes to artificial infant formula [67].

## 8. Genetics, Epigenetics, and Raw Cow’s Milk Exposure

Allergic diseases and asthma have a strong genetic background. It has been shown that environmental factors may interact with the genome both by modifying the environmental effect by the underlying genome or, vice versa, the genetic effect by environmental exposure. As was presented above, polymorphisms in the CD14 receptor influenced a protective effect of unpasteurized milk consumption on allergy and asthma [61]. In a genome-wide association study in the GABRIEL population, the gene–environment interaction for atopy, asthma, and farming exposures were tested. The common genetic polymorphisms previously described in asthma did not modify the protective effect of farming exposures on asthma, although the interaction was detected in rare SNPs [68]. The exposure to raw farm milk in pregnancy and the first year of life was associated with changes in the gene expression of innate immunity receptors [69]. Recent epigenetic studies of genomic adaptation to the environment that was not caused by changes in the nucleotide sequence of the genetic code itself revealed additional methods of interaction. Large-scale genome-wide meta-analysis of DNA methylation and childhood asthma identified novel epigenetic variations, which might be potential biomarkers of later asthma risk [70,71]. In a prospective PASTURE study, DNA methylation patterns changed significantly in the first year of life in asthma-related genes, and exposure to farm environments seemed to influence methylation patterns in this population [72]. Further studies on the effects of raw milk exposure as well as genome and epigenome interactions on asthma and allergy phenotypes are needed to disentangle these complicated associations.

## 9. Conclusions

There is a debate about the role of raw cow’s milk role in human health. Sceptics say that raw milk carries a significant risk of bacterial pathogens infection and there is no clear evidence that raw milk has any nutritional benefits compared to pasteurized milk. Enthusiasts see in milk the hope for effective prevention of allergic diseases and even respiratory tract infections [73]. There is no doubt that the components of raw milk can influence the immune function, but the final proof based on controlled studies in infants is not possible due to ethical reasons. Undoubtedly, even if the final understanding of the role of raw cow’s milk seems to be a distant prospect, it is one of the most intriguing and promising paths to be studied in allergy prevention.

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
