# Peer review of "Raw Cow’s Milk and Its Protective Effect on Allergies and Asthma"

_nutrients, 2019, doi:10.3390/nu11020469_

Round 1
Reviewer 1 Report
(nutrients-449867)
The manuscript entitled “Raw cow’s milk and its protective effect on allergies and asthma”, by Barbara Sozanska, describes a well written review based on previously reports on the possible protective effects of raw cow’s milk and prevalence and susceptibility atopic disease, allergies and Asthma. This manuscript not only focus on previously reported association of farm and processed milk and disease prevalence’s per se, but also takes the discussion further towards plausible underlying molecular mechanism that may contribute to disease. With respect to the latter, several distinct topics pass the revue, e.g. the differences between farm and processed milk in relation to the content of (intact) milk proteins, fats or fatty acids. Subsequently, this manuscript describes the possible effects or interaction with disease and the microbiome within farm milk and processed milk, and also the interaction of milk contents with the microbiome of the consumer in that respect. Finally this manuscript briefly touches the topic of possible underlying epigenetic mechanisms (via exosomal miRNA’s). With respect to the topic (title) of this manuscript, I find this study very complete and certainly suitable for publication in Nutrients.
I have however one minor point. Allergies in general, and certainly asthma, have been shown to involve a considerable explaining genetic component. Moreover, as this review describes, many environmental factors, that apparently play a large role in the manifestation of disease, might affect the human epigenome. In that context the authors may consider to deepen the genetic and epigenetic interacting element in this review slightly more. The latter by describing and discussing the fact that these components may contribute directly or indirectly to disease expression, while raw cow’s milk possible does not activate certain immunological pathways that promote disease despite a genetic susceptibility.
Author Response
It is my pleasure to submit the revised manuscript entitled “Raw cow’s milk and its protective effect on allergies and asthma” for publication in a special issue “Cow’s milk and Allergy” of Nutrients. As before, the manuscript has never been published and is not under consideration for publication elsewhere.
Generally, I would like to thank the reviewer for his valuable comments. I am confident that they helped us to improve the manuscript. Please find the specific responses below. I have uploaded file with all changes tracked.
REVIEWER: I have however one minor point. Allergies in general, and certainly asthma, have been shown to involve a considerable explaining genetic component. Moreover, as this review describes, many environmental factors, that apparently play a large role in the manifestation of disease, might affect the human epigenome. In that context the authors may consider to deepen the genetic and epigenetic interacting element in this review slightly more. The latter by describing and discussing the fact that these components may contribute directly or indirectly to disease expression, while raw cow’s milk possible does not activate certain immunological pathways that promote disease despite a genetic susceptibility.
Author Response
Thank you very much for this valuable comment. I have added a short, separate paragraph (lines 288-306) dedicated to recent epigenetic studies in the field.
Reviewer 2 Report
The manuscript reviews the protected effect of raw cow’s milk on allergies and asthma. It tracks epidemiological studies on this topic, elaborates on the difference between raw and commercial milk. The manuscript also studies links between allergies and asthma and the raw milk protein, fatty acids, microbial and exosomes content.
The review covers and important topic particularly with significant increase in prevalence of allergic diseases and its association with the hygiene hypothesis. The manuscript managed to give a good idea on the protective role of raw cow’s milk on allergies and asthma by appropriately citing generally recent 69 publications, only 2 of which are self-citations. The references include important references in the area like The GABRIELA study by Loss et al (2011) and most recent Abbring et al (2019) article. As the review lists relevant results on the topic it also provides appropriate synthesis and connection.
Although the overall write-up is appropriate the author should consider paraphrasing some text: page 7 lines 226-232 and page 8 lines 266-275
Author Response
I would like to thank the reviewer for his valuable comments. I am confident that they helped us to improve the manuscript. Please find the specific responses below. I have uploaded file with all changes tracked.
REWIEVER: Although the overall write-up is appropriate the author should consider paraphrasing some text: page 7 lines 226-232 and page 8 lines 266-275
Author Response
According to the Reviewer suggestion, I have paraphrased indicated text to become more clear for the reader.